# Looking at microbial metabolism by high-resolution $^2$H-NMR spectroscopy

Victor P. Kutyshenko[1], Petr M. Beskaravayny[1], Maxim V. Molchanov[1], Svetlana I. Paskevich[1], Dmitry A. Prokhorov[1] and Vladimir N. Uversky[2,3]

[1] Institute of Theoretical and Experimental Biophysics, Russian Academy of Sciences, Pushchino, Russia

[2] Institute for Biological Instrumentation, Russian Academy of Sciences, Pushchino, Moscow Region, Russia

[3] Department of Molecular Medicine and USF Health Byrd Alzheimer's Research Institute, College of Medicine, University of South Florida, Tampa, Florida, USA

## ABSTRACT

We analyzed the applicability of high-resolution $^2$H-HMR spectroscopy for the analysis of microbe metabolism in samples of mitochondrion isolated from rat liver and from aqueous extracts of homogenates of rat liver and other organs and tissues in the presence of high $D_2O$ contents. Such analysis is possible due to the fast microbe adaptation to life in the heavy water. It is also shown that some enzymatic processes typical for the intact cells are preserved in the homogenized tissue preparations. The microbial and cellular metabolic processes can be differentiated via the strategic use of cell poisons and antibiotics.

Corresponding authors
Victor P. Kutyshenko,
kutyshenko@rambler.ru
Vladimir N. Uversky,
vuversky@health.usf.edu

## INTRODUCTION

Recent years witnessed an increased interest of researchers in the analysis of various biological fluids. This research is taken now as a fundamental basis of metabolomics which studies the metabolic profiles of animals and humans during their normal activity and at various pathological conditions, as well as looks at the effects of various drugs and other substances on specific organ/tissue, the whole organisms, and even on the entire ecosystem (*Holmes, Wilson & Nicholson, 2008*; *Maher et al., 2008*; *Nicholson & Lindon, 2008*). Typically, the term 'biological fluids' is taken as a synonym to 'body fluids' or 'biofluids' that correspond to liquids originating from inside the bodies of living people, such as urine, blood, saliva, sweat, cerebrospinal fluid, mucus, etc. However, this concept can be extended to include water washouts and aqueous extracts of the homogenates of various organs and tissues of animals (*Kutyshenko et al., 2007*; *Kutyshenko et al., 2008a*; *Kutyshenko et al., 2008b*) and plants (*Molchanov et al., 2012*). Addition of these somewhat artificial biological fluids leads to the noticeable increase in the variability of experimental material suitable for comprehensive analysis and produces substantial information related not only to the organs under study, but also to the interactions of these organs with the remaining organism and with specific microorganisms.

The close connection between plants and animals with specific microorganisms constituting microbiomes or microbiotas is a well-established fact. In fact, animals, including humans, constantly coexist with microorganisms, being involved in numerous symbiotic interactions with various bacteria and yeast that densely populate intestines, skin, and tunica mucosa of airways, pharynx and urinary tract. Furthermore, some microorganisms can get access to various organs through bloodstream or other biofluids leading to the development of various pathologies. The current list of human symbiotic microorganisms includes ∼5,000 species that are uniquely distributed between 15 and 18 sites of permanent habitat in males and females, respectively (*Human et al., 2012a*; *Human et al., 2012b*). Since different organs are biochemically different, sets of symbiotic microorganisms populating them can vary between different organs within the same organism. Many members of the human microbiome are conditionally pathogenic microorganisms that can provoke development of various maladies if appropriate conditions are given (*Tancrede, 1992*; *Riabichenko & Bondarenko, 2007*; *Yu et al., 2012*). Under these circumstances, originally harmless and even beneficiary symbiotic microorganisms can go bad and start negatively affect the normal cellular and organ functions of the host organism, secreting specific toxins and ferments and eventually leading to the metabolism distortion and cell death. Furthermore, by destroying the host cells, microorganisms promote the release of the cell content into the extracellular environment, thereby further exacerbating the course of a disease and negatively affecting the overall condition of the host organism. In fact, sometimes, massive cell death can be a self-propagating process, where proteins released from the dying cells affect neighboring cells leading to their death and consequently generating favorable conditions for the propagation of both the "own" symbiotic microorganisms of the microbiome and the microorganisms introduced from the outside. Therefore, under such circumstances, therapy should include both antibacterial and healing strategies.

In this work, the mitochondria suspension and the aqueous extracts from the liver homogenates are used to model cell death and organ damage (necrosis) resulting from the injuries and pathologies and to experimentally characterize the related processes. We propose here an instrumental approach that can be used to detect and control both microbial and host enzymatic processes taking place within the sites of disease origin. This approach is based on the detection of the deuterium incorporation to the specific metabolism products. Here, deuterium (in the form of heavy water) is added directly to the medium where the fermenting and/or microorganism vital activity takes place. Our earlier analysis revealed that many microorganisms can easily adapt to the conditions of high heavy water contents, and the presence of almost 100% heavy water does not significantly affect normal functioning of certain microorganisms (*Kushner, Baker & Dunstall, 1999*; *Molchanov et al., 2012*). Under these conditions, deuterium can be incorporated to the substrates due to the existence of efficient exchange between the protons of organic moieties of substrates and deuterium present in media. Next, these deuterated substrates can be used in biochemical reactions leading to the enzymatic incorporation of deuterium to

the corresponding metabolism products (*Ewy, Ackerman & Balaban, 1988*; *Kushner, Baker & Dunstall, 1999*; *Budantsev, Uversky & Kutyshenko, 2010*; *Molchanov et al., 2012*). One of the most informative techniques to follow the mentioned processes in biological fluids is the high-resolution NMR at the deuterium nuclei, $^2$H-NMR (*Budantsev, Uversky & Kutyshenko, 2010*). In comparison with proton spectra, $^2$H-NMR spectra are characterized by lower resolution and lower sensitivity. Furthermore, deuterium-deuterium couplings are about 40 times smaller than proton-proton couplings deuterium-deuterium couplings. However, the overall shapes of $^1$H-NMR and $^2$H-NMR spectra of organic components are rather similar. The only exception here is the fact that due to the low spin-spin interaction constants, quadrupole broadening, and the presence of various isotopomers, the multiplets seeing in the $^1$H-NMR spectra are typically presented by broad 'singlets' in the $^2$H-NMR spectra (*Emsley, Feeney & Sutcliffe, 1966*). However, despite the aforementioned issues, $^2$H-NMR spectroscopy has numerous advantages.

In this work, we show the applicability of the high-resolution $^2$H-NMR spectroscopy for the quantitative analysis of biological fluids using preparations of mitochondria suspension and aqueous extracts from rat liver homogenates as illustrative example. It is important to emphasize here that the proposed approach for studying microbial and host enzymatic activities based on the analysis of deuterium incorporation to the metabolic products can be of wide practical use in many other cases, when high $D_2O$ concentrations do not perturb the physiological processes of the studied (*Budantsev, Uversky & Kutyshenko, 2010*).

## MATERIALS AND METHODS

Mitochondria were isolated from the livers of Wistar rats using the standard protocols (*Belosludtsev et al., 2009*). Mitochondria samples used in our study were a generous gift of Prof. Mironova GD. The only modification of the isolation protocol in some preparations was substitution of light water by heavy water (OOO Astrochim, Russia, 99.8%) done at our request. The standard functional analysis revealed that the mitochondria isolated using such modified heavy water-based protocol were active and preserved their activity for several hours after isolation. Part of mitochondria isolated by a standard, light water-based approach was subsequently treated with heavy water. The heavy water content in samples was controlled using the characteristic features of $^1$H-NMR spectra. On average, the heavy water content ranged from 40% to 57% in various samples prepared using the light water-based approach and was higher than 86% in samples prepared by heavy water-based approach. The freshly prepared samples had pH $\sim$ 7.

Livers of the Vistar rats were a kind gift of Prof. Kichigina VF. These animals were sacrificed for the purpose of unrelated experiments (*Popova, Sinelnikova & Kitchigina, 2008*). Aqueous extracts of the rat liver homogenates were prepared using $0.40 \pm 0.03$ g samples which were first carefully homogenized in the eppendorfs using a special sterile glass spatula and then diluted with 0.75 ml heavy water (CIL, USA, 99.9%). Samples were centrifuged using the microcentrifuge CM-50 (ELMI, USA) prior to the NMR measurements. Measurements were taken one day after sample preparation. Freshly

prepared samples contained 60% heavy water and had pH 6.3-6.1. With time, the medium was moderately acidified (to pH $\sim$ 5.0) due to the lactate formation. Although sample preparation was carried out carefully and thoroughly, no special steps were taken to ensure sample sterility.

Antibiotics gentomicin (Asparin, Germany) and amphotolecin B (Sigma) used for prevention of microbial contamination at the cell culture (*Solovieva et al., 2008*) were dissolved in 2 ml of $D_2O$ to ensure final concentrations of 40 μg/ml (gentamicin) and 2 μg/ml (amphotolecin B) in the mitochondria suspension samples and of 5 μg/ml (gentamicin) and 0.4 μg/ml (amphotolecin B) in the liver homogenate samples. Sodium azide concentration was kept at the level of 0.2%.

NMR spectrometer AVANCE 600 (BRUKER) with the operating frequency 600.13 MHz was used in the experiments. $^1$H-NMR spectra were measured using the spectral width of 8000 Hz, 90° pulse of 11 μs, and temperature of 298 K. As a rule, 128 accumulations were sufficient to obtain good signal to noise ratio. The NMR spectra were obtained using the pulse sequences "WATERGATE" and "ZPRG" with the relaxation delay from 1 to 3 s. The heavy water content was determined using the "ZG" pulse sequence. Here, NMR spectra of the samples with known ratios of light and heavy water were measured with the relaxation delay of 10 s. These spectra were analyzed to measure the spectral intensities of water signal which then were used to make a calibration plot. Heavy water content in all working samples was evaluated using this calibration plot.

$^2$H-NMR spectra were measured using the 20 W field stabilizer at the frequency of 92.12 MHz, 90°-impulse length of 150 microseconds, a spectrum width of 8000 Hz and 500–1000 accumulations. All the measurements were taken at 298 K inside the sensor. The Fourier transformation was carried out at doubled point array using the simple exponential multiplication with the constant of 1.0 Hz and 0.2 Hz for the proton and deuterium spectra, respectively.

## RESULTS AND DISCUSSION

### Mitochondria from rat liver

It is believed that the mitochondria isolated from the rat liver preserve their functional activity *in vitro* for 1–3 h after isolation (*Belosludtsev et al., 2009*). The proton NMR spectrum of the suspension of mitochondria isolated using the heavy water-based protocol that was collected during this initial time of the sustained mitochondrial activity is shown in Fig. 1A. This spectrum is dominated by the rather broad signals typical of the intracellular organic molecules. Note that narrow and very intensive signals correspond in a region from 4.7 to 3.5 ppm to sucrose, which is present in the extracellular medium due to the peculiarities of the isolation protocols (Fig. 1A) (*Belosludtsev et al., 2009*). In the absorption region of the aliphatic protons (from 3.0 to 0.5 ppm), the major components are broad signals corresponding to the mitochondrial membranes. After 10–12 h of incubation, some sharp signals start to appear (see Fig. 1B). These signals correspond to the organic molecules extruded from the mitochondria to the medium. With time, the amplitudes and number of these sharp signals increase, whereas the amplitudes of
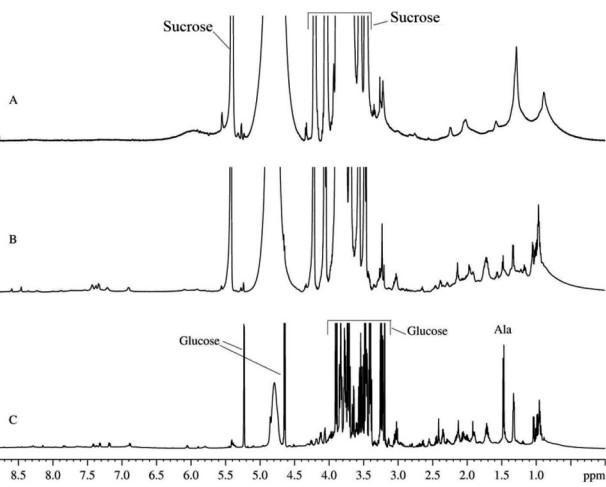

**Figure 1** **¹H-NMR spectra of biological fluids.** (A) Mitochondrion isolated from rat liver. Measurements were taken immediately after mitochondrion isolation. (B) Mitochondrion isolated from rat liver. Measurements were taken one day after isolation. (C) Aqueous extract of the rat liver homogenate.

broad signals proportionally decrease. At this moment, spectrum contains signals of free amino acids and other organic components, which are commonly detected in other biological fluids and aqueous extracts from various plant and animal tissues. Figure 1C shows typical ¹H-NMR spectrum of the aqueous extracts of the rat liver homogenate. Spectrum contains sharp signals of free amino acids that coincide with signals detected in all major biological fluids. In fact, ¹H-NMR spectra of the biological fluids studied so far are quantitatively similar, possessing some fluid/condition-specific qualitative differences. Comparison of Figs. 1C and 1B revealed that the majority of sharp signals detected in the ¹H-NMR spectrum of the aqueous extract of the rat liver homogenate coincide with those in the ¹H-NMR spectrum of the mitochondria. In the ¹H-NMR spectrum of the aqueous extract of the rat liver homogenate, the most characteristic signals with highest intensities correspond to glucose. During the observation for 3–5 days, proton spectra of the aqueous extracts did not change neither qualitatively nor quantitatively.

Interestingly, signals in the ²H-NMR spectrum with the satisfactory signal-to-noise ratio that can be used for the qualitative measurements start to appear only after the incubation for about 20 h, although some signals are clearly detectable at earlier time points. On a second day, the ²H-NMR spectrum is completely formed, and subsequent incubation results in the increase of amplitudes of already existing signals. Figure 2 represents this process by showing normalized integral intensities measured in the range of 3.6–0.0 ppm of proton spectra (black circles) or in the range of 4.2–0.0 ppm of ²H-NMR spectra. The increase in the amplitudes of sharp signals in the proton spectra is related to the gradual release of the intramitochondrial organic compounds resulting from the destruction of mitochondrial membranes.

During the first 27 hours after isolation of mitochondria, the kinetics of the formation of proton- and deuterium-containing metabolites are similar due to the insignificant amounts of the low molecular mass (LMM) compounds released from the destroyed
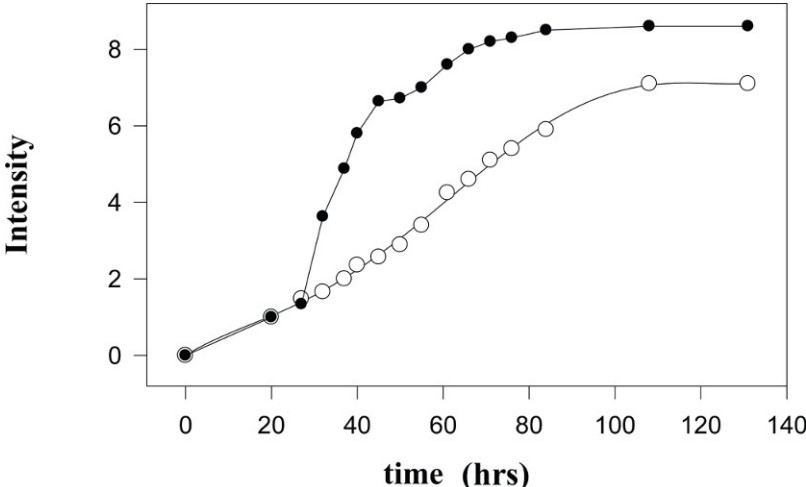

**Figure 2 Kinetic changes in the integral intensities of the NMR spectra.** Time courses of changes in the integral intensities of the aliphatic part of [1]H-NMR (black circles) and [2]H-NMR spectra (open circles).

mitochondria. These LMM compounds serve as substrates for the metabolism of the contaminating microorganisms and for the residual enzymatic activity of the mitochondrial proteins either released to the medium from the destroyed mitochondria or still located inside the damaged mitochondria. At longer incubation times, kinetic parameters of the observed processes become more and more different. This reflects the existence of an active metabolic conversion of the released substrates by microorganisms and by the residual enzymatic activity of mitochondria. Importantly, the proton spectra of mitochondria do not qualitatively change with time; i.e., no new signals appear and no old signals completely disappear. The sharp increase in the intensity of signals in the [1]H-NMR spectra at the beginning of the second day is associated with the massive death of mitochondria. Exponentially slowing, this process continues for some 50 hours. A plateau and subsequent small increase in the vicinity of 50 hours are determined either by the death of the least sensitive cells or by the 'switching on' of some other degradation mechanisms. The monotonous increase in the signal intensity of the [2]H-NMR spectra is associated with the enzymatic activity and the microbial metabolism. On average, the integral intensities of the [2]H-NMR spectra are about 1.3-times lower than the amplitudes of peaks in the proton spectra.

Figures 3A and 3B represent a pair of typical [2]H-NMR spectra measured for two mitochondrial isolates randomly selected from a dozen of independent isolation performed during a year using different isolation protocols (sucrose-based and mannitol-sucrose-based), on the basis of $D_2O$ and $H_2O$, respectively. All the recorded spectra possess close similarity to each other, mostly differing in relative intensities of several peaks. Figure 3 represents signal assignments based on the comparison of chemical shifts with proton spectra of known metabolites from various biological fluids. These assignments took into account the presence of the isotope shift and were performed using a large set of [2]H-NMR spectra of samples prepared from various plant and animal sources. The major difference

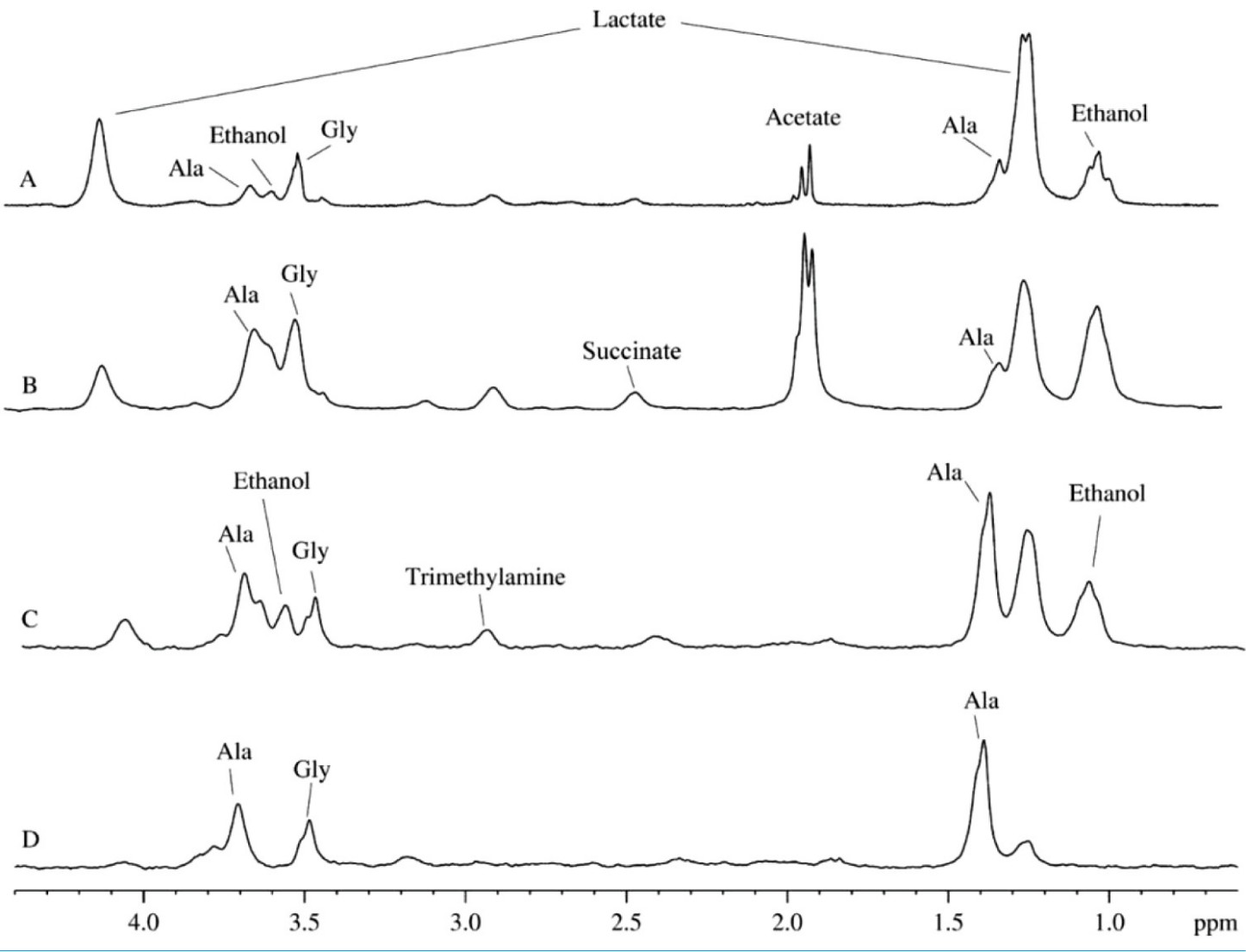

**Figure 3** **$^2$H-NMR spectra of biological fluids.** (A) Mitochondrion isolated using $D_2O$-based protocol. (B) Mitochondrion isolated using D2O-based protocol. (C) Aqueous extract of the rat liver homogenate. (D) Aqueous extract of the rat liver homogenate with sodium azide added.

between spectra shown in Figs. 3A and 3B is in lesser amounts of ethanol and acetate in mitochondrial preparations utilizing heavy water. Furthermore, in all the cases of heavy water-based isolations, the rightmost signal corresponding to isotopic variant of acetate ($-CD_3$) was always higher than the middle signal corresponding to $-CHD_2$, since the heavy water content in these samples was $\sim$85%, whereas in light water-based isolations with concomitant addition of $D_2O$, the heavy water content was at the level of 35–40%. The presence of signals corresponding to ethanol, acetate and formate at 8.43 ppm (not shown) is the reflection of the microbial contamination of the isolated mitochondria.

Figure 3C represents a typical $^2$H-spectrum of the aqueous extract of liver homogenate. This spectrum, being corrected for the differences in intensity of some signals, resembles the spectrum of the mitochondria isolates. However, since this spectrum possesses signals

corresponding to ethanol, formate, and acetate, one can suggest that these samples were contaminated by microorgansims. To identify signals corresponding to the products of the microbial metabolism, some broad-spectrum antibiotics or sodium azide were added during the sample preparation. Similar to antibiotics, sodium azide (low concentrations of which are used as preservatives in the food industry) possess antimicrobial activities. Sodium azide predominantly affects Gram-negative bacteria, suppressing their growth and development. The application of both bactericides had similar outputs, and the resulting $^2$H-NMR spectra of the aqueous extract of liver homogenates treated with antibiotics and sodium azide were identical.

Figure 3D represents one of the spectra for bactericide-treated sample and shows the lack of signals corresponding to ethanol, formate, and acetate, supporting their bacterial origin. Therefore, resulting spectra contain only signals corresponding to the compounds produced by mitochondrial enzymes under the proton-deuterium exchange conditions. The liver extracts contain both substrates and ferments that participate in the enzymatic reactions uncontrolled by the decomposed cells. The corresponding $^2$H-NMR spectra contain alanine, glycine, and lactate (Fig. 3D), with alanine being the dominating component. It is known that alanine accounts for ∼30% of all amino acids delivered to the liver. This explains relatively high concentrations of alanine in the liver preparations (see Fig. 2C). In the liver, alanine is converted to pyruvate, which is subsequently used for the glucose synthesis (*Malaisse et al., 1996*; *Burelle et al., 2000*).

In our experiments, the samples were prepared by the mechanical homogenization of rat livers. Therefore, the resulting homogenate contains some surviving cells that remain functional and continue function more-or-less normally, at least for some time. Therefore, these preparations can be considered as a model of severe tissue damage. Survived cells continue to express proteins and possess metabolic processes supporting cell life activity. Under the oxygen deficiency conditions of our experiments, the only available pathway for energy generation in a cell is anaerobic glycolysis. However, the last stage of this pathway is likely to fail as evidenced by the lack of the increase in the lactate signal in the corresponding $^1$H-NMR spectra (see Fig. 1C).

Pyruvate produced during glycolysis is converted to the alanine via the transamination reaction. This reaction together with the reversed transformation of alanine to pyruvate is catalyzed by the alanine transaminase also known as alanine aminotransferase (*Dolle, 2000*; *Yang et al., 2009*). The activity of this enzyme combined with the protein degradation and membrane decomposition, together with the presence of some free alanine inside the cells give likely explanation for the moderate increase in the alanine signal in the spectra of rat liver homogenates during their long-term observation. The presence of deuterium in the Cα position and in the methyl groups of alanine supports the enzymatic origin of alanine's hydrocarbon skeleton (see Fig. 4).

Figure 5 represents the $^2$H-NMR spectra of mitochondria in samples containing antibiotics. Comparison of spectra measured at different time points after the sample preparation indicates the presence of some kinetic processes. Figure 5C shows signals accumulated during the first 8 hours of sample incubation. The most intensive signal

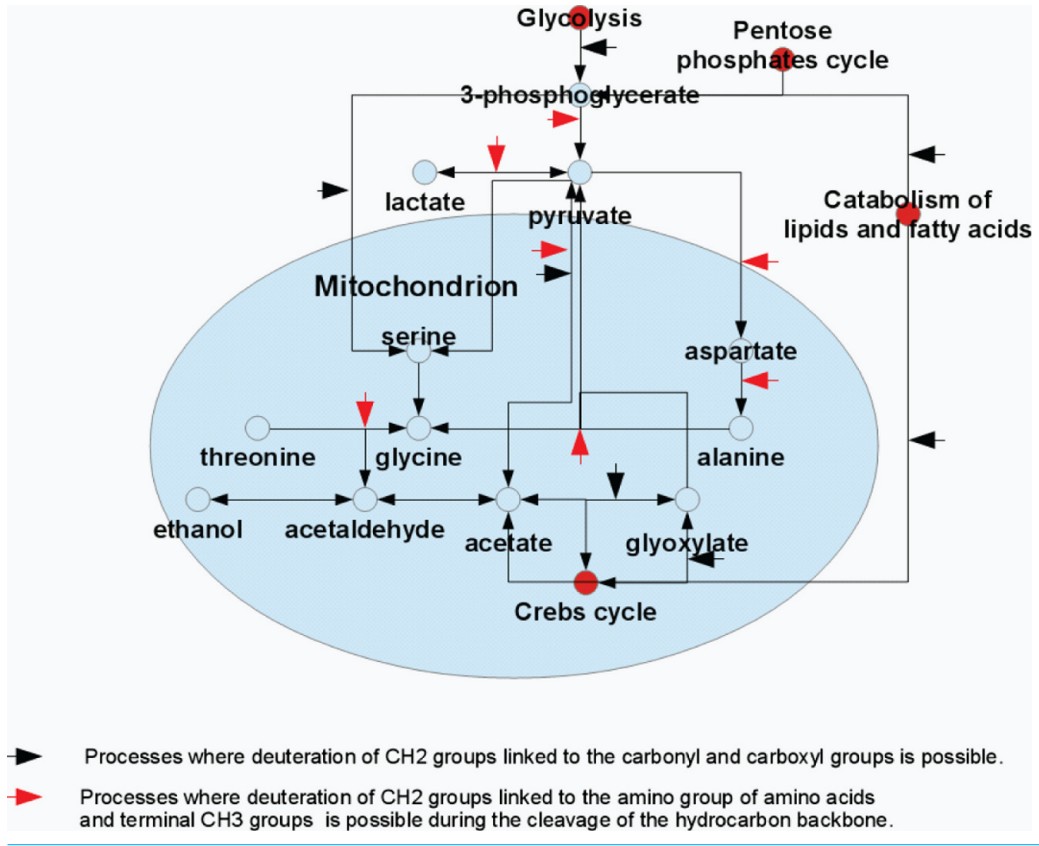

Processes where deuteration of CH2 groups linked to the carbonyl and carboxyl groups is possible.

Processes where deuteration of CH2 groups linked to the amino group of amino acids and terminal CH3 groups is possible during the cleavage of the hydrocarbon backbone.

**Figure 4 Model representation of the metabolite conversion pathways.** Various pathways of the metabolite conversion in cytosol and mitochondrion of rat liver at which hydrocarbon skeleton of resulting compounds can be deuterated.

here is a signal from the glycine deuterons followed by a less intensive signal of deuterated alanine. Furthermore, the spectrum contains signals corresponding to the proton-deuterium exchange at nitrogens of urea (5.7 ppm), and side chains of glutamine (∼7.6 ppm) or asparagine (∼6.9 ppm) or both residues (∼7.6 ppm and ∼6.9 ppm). These signals significantly increase after one day of incubation (see Fig. 5C) but did not change much during the more prolonged incubation.

To the sixth day, the spectrum undergoes further changes, and signals of lactate and formic acid appear, whereas signals corresponding to the nitrogen disappear. These changes reflect starting bacterial activity leading to the nitrogen utilization and appearance of own metabolites. Concentrations and ratios of antibiotics were carefully selected to suppress the bacterial activity and not to produce additional damage of the liver cells. In these settings, the bacterial activity was sufficiently suppressed, since in the absence of antibiotics, signals corresponding to lactate and ethanol were easily detectable after only 2–3 days (see Fig. 3).

The major glycine biosynthetic pathway in a cell is the one catalyzed by the serine hydroxymethyltransferase, an enzyme that plays an important role in cellular one-carbon pathways by catalyzing the reversible, simultaneous conversions of L-serine to glycine

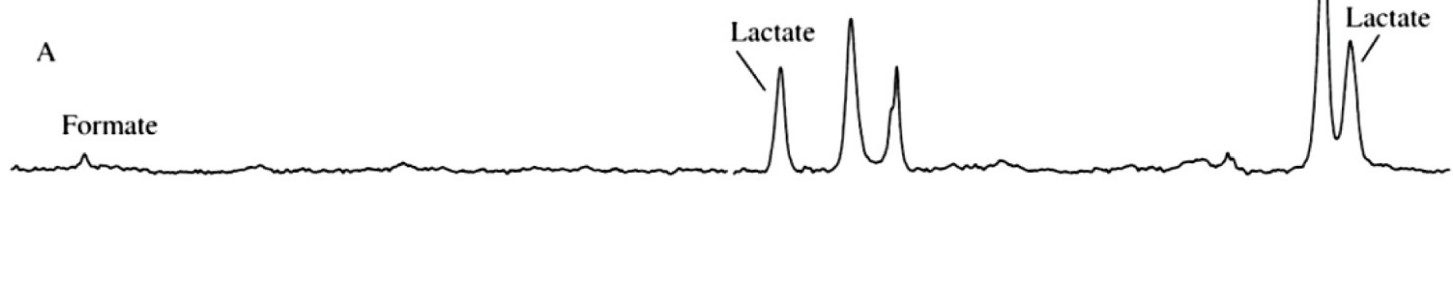

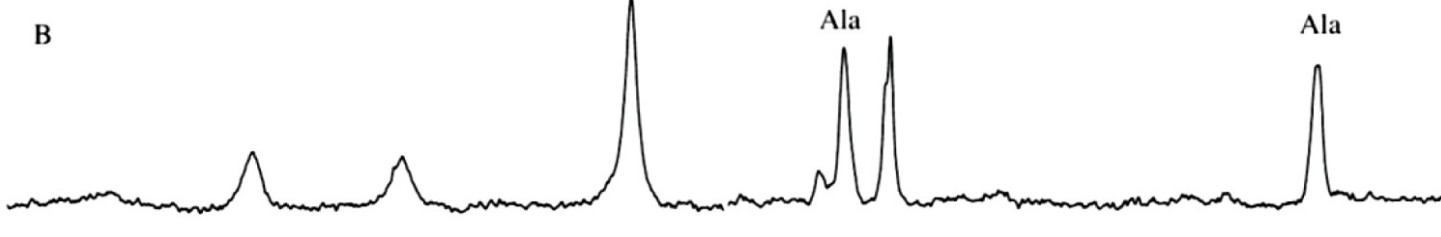

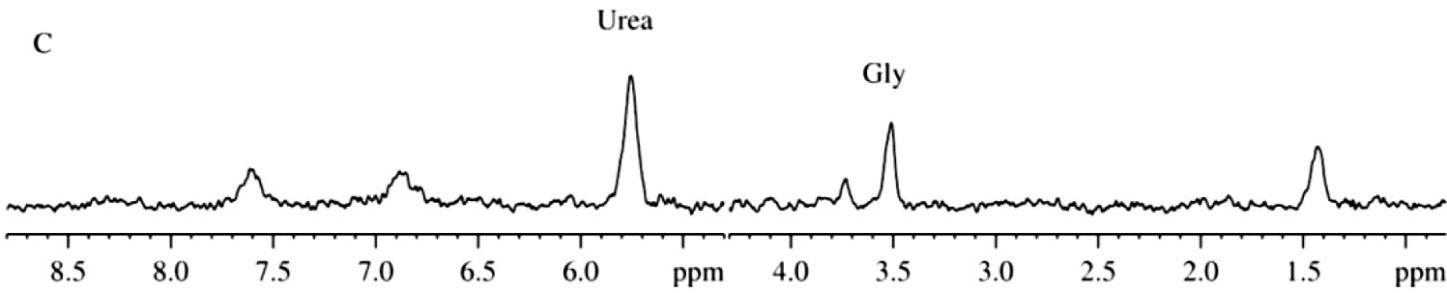

**Figure 5 Time course of changes in the $^2$H-NMR spectra of mitochondrion samples with added antibiotics.** (A) Spectrum is taken on the sixth day after the sample preparation. (B) Spectrum is taken on the second day after the sample preparation. (C) Spectrum is taken 8 h after the sample preparation.

(retro-aldol cleavage) and tetrahydrofolate to 5,10-methylenetetrahydrofolate (hydrolysis) (*Appaji Rao et al., 2003*; *Scheer, Mackey & Gregory, 2005*; *Berdyshev et al., 2011*). Figure 4 shows that serine is synthesized in a cell from the 3-phosphoglycerate, which is one of the intermediates of the glycolysis, and glutamine, which serves as the source of amine. Serine is subsequently used for the protein biosynthesis and for the synthesis of phosphatidylserine that constitutes typically ∼15% of all membrane phospholipids. The transfer of the serine methyl group to tetrahydrofolate in the presence of heavy water can be accompanied by the deuteration of the $CH_2$-group of the newly synthesized glycine.

Our study revealed that high-resolution $^2$H-NMR spectroscopy can be successfully used in metabolomics studies. Furthermore, the strategic use of antibiotics helps discriminate

microbial activity from enzymatic cellular processes. The major products of microbial activity are organic acids, such as formate, acetate, lactate, propionate (seeing in spectra of homogenates of heart muscle) and ethanol. It is important to note here that our data suggests that ethanol can originate not only from the classical alcoholic fermentation but can be generated via other processes. This conclusion is based on the uneven intensities of $-CD_2-$ and $-CD_3$ deuterons reproducible detected in our experiments, whereas these signals would have comparable intensities if ethanol was exclusively generated via the alcoholic fermentation pathway due to the more efficient deuteration of the methylene group (*Kutyshenko & Iurkevich, 2000*).

The major substrates for the ethanol formation are pyruvic acid and acetaldehyde. There are several biosynthetic pathways for the production of these compounds in the organism, and pyruvate and acetaldehyde can be generated from glucose (as a result of glycolysis), pentoses (via pentose phosphate pathway) or from some amino acids (e.g., due to the catabolism of alanine and threonione) (see Fig. 4). Therefore, the ethanol formation is likely a reflection of the successful development of the contaminating bacterial and fungal microbiomes. Based on the characteristic patterns of the hydrogen substitution by deuterium, we hypothesize that the significant part of the endogenous ethanol in our settings is synthesized from the deaminated amino acids (see Fig. 4). For example, during the processes of alanine transamidation and threonine degradation, the resulting terminal $CH_3$-groups of pyruvate and acetaldehyde are efficiently deuterated. The subsequent fermentation of pyruvate to ethanol in the presence of heavy water may be accompanied by the deuteration of ethanol's $-CH_2$-group. Resulting $^2$H-NMR spectra of ethanol derived from these intermediates suggest an almost proportional saturation of $CH_3$- and $-CH_2$-groups, in sharp contrast to the disproportional saturation of these groups in ethanol molecules produced via the glucose fermentation.

## ACKNOWLEDGEMENTS

We are thankful to Professors Kichigina VF, Mironova GD and Dr. Popova I for providing biological samples used in this study.

### Funding

This work was supported by a grant from Russian Foundation for basic research (10-02-00996) and the Leading Scientific Schools program (850.2012.4). The funders had no role in study design, data collection and analysis, decision to publish, or preparation of the manuscript.

### Grant Disclosures

The following grant information was disclosed by the authors:
Russian Foundation for basic research: 10-02-00996.
The Leading Scientific Schools program: 850.2012.4.

## Competing Interests

Vladimir N. Uversky is an Academic Editor for PeerJ. There are no other competing interests.

## Author Contributions

- Victor P. Kutyshenko conceived and designed the experiments, analyzed the data, wrote the paper.
- Petr M. Beskaravayny, Maxim V. Molchanov and Svetlana I. Paskevich performed the experiments, analyzed the data.
- Dmitry A. Prokhorov conceived and designed the experiments, performed the experiments, analyzed the data, wrote the paper.
- Vladimir N. Uversky analyzed the data.

## Supplemental Information

Supplemental information for this article can be found online at http://dx.doi.org/10.7717/peerj.101.

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
