# Peer review of "Looking at microbial metabolism by high-resolution 2H-NMR spectroscopy"

_PeerJ, doi:10.7717/peerj.101_

## Round 0.1 · original submission · Minor Revisions

Dear Dr. Uversky,
Woud you like to answer the reviewers questions and correct your manuscript according to the reviewers criticism.

Reviewer 1 ·

Basic reporting

Spell checking needed.
such as page 3, line 50, some not "come', line 125, 90 pulse, not "impulse".

Experimental design

More detailed information of sample preparation and NMR experiments should be providied. What is the heavy water conentent in the homogenates sample?
What is the concentration of antibiotics? how to make sure the concentration is enough to suppress the bacterial activity and not affect the metaboic of liver cells?
What pulse sequence used to collect 1H and 2H spectra? 2H spectra were acquired in lock channel? How long is relaxation delay (d1) used? no water suppression? NMR signals processing parameters.

Validity of the findings

In fig.3 and fig5, the 2H spectra of samples treated with sodium azide and antibiotics show differences, what casue the differences? Spectra window should be the same in fig 3 and 5.
It seems huge peak of D2O was not shown in the spectra, the author should state it clearly in the figure lengend or in the text.

Additional comments

The advatage and necessary to use insensitive 2H spectra should be given in the introduction.
The assignments of the peaks shound cite proper references.

Reviewer 2 ·

Basic reporting

In the manuscript “Looking at microbial metabolism by high-resolution 2H-NMR spectroscopy” by Kutyshenko et al., a new approach for examination of the biological fluids is developed.
It is shown that high-resolution 2H-NMR spectroscopy is an effective tool for the quantitative analysis of aqueous extracts of the rat liver homogenates. It was also demonstrated that cellular metabolic processes can be differentiated via cell poisons and antibiotics.
It is concluded that this approach applicable for other cells and tissues become an effective tool for detection and control of both microbial and host enzymatic processes.

I have several minor comments:
1. page 2, line 21, page 4, lines 71-72: Abstract and Introduction it is announced that the work will be done on samples of mitochondrion isolated from rat liver and from aqueous
extracts of homogenates of rat liver and other organs and tissues, but in reality data only for mitochondrion isolated from rat liver and from aqueous extracts of homogenates of rat liver are presented.

2. page 4, lines 65-69:
It is not clear the propagation of what? death?

3. page 5, lines 89-96
The phrase (lines 92- 96) is too long and confused. It is not clear whether the lower resolution and lower sensitivity of 2H-NMR is advantage or disadvantage?
Misprints must be eliminated.

4. Figure 1 represents the transformation of the 1H-NMR spectrum of mitochondrion isolated from rat liver isolated from rat liver and that of aqueous extract of the rat liver homogenate.
There are two questions 1) what is the difference in 1H-NMR spectra between mitochondrion from different isolations and 2) when 1H-NMR spectrum of aqueous extract was recorded – immediately after preparation, or one day after?

5. Several comments to Figure 2.
1) Figure 2 shows that for about 30 hrs the amplitudes of 1H-NMR and 2H-NMR coincide.
1H-NMR spectrum immediately after preparation is given in figure 1. On the page 8, lines 159-160, it is said that signals in the 2H-NMR spectrum start to appear only after incubation for
about 24 hrs. After another day of incubation, the 2H-NMR spectrum is completely formed. Interestingly, what it represents immediately after sample preparation?
But on the same page below (lines 173-174) it is said: “During the first 27 hrs after isolation of mitochondrion, the kinetics of the formation of proton- and deuterium-containing metabolites are similar due to the insignificant amounts of the low molecular mass (LMM) compounds released from the destroyed mitochondria.”
2) The samples that were used for constructing Figure 2 must be indicated in figure legend.
3) How the authors explain shoulder at about 50 hrs on the 1H-NMR spectrum?

6. Figure 3A and 3B. The identical figure legends puzzle the reader. The difference is explained only on the page 9, lines 185-188. It must be given in figure legend.
It would be good to indicate when these spectra were recorded in time scale after samples preparation.

7. Figure 3 and Figure 5 shows 2H-NMR spectra of the aqueous extract of liver homogenates treated with sodium azide and antibiotics. It is claimed that the action of sodium azide and antibiotics is the same. It is not clear when 2H-NMR spectra presented in Figure 3 D was recoded. To what spectrum in Figure 5 it corresponds? How the author could comment the difference of spectra presented in Figure 3D and 5?

Experimental design

Good

Validity of the findings

New approach for examination of the biological fluids is developed.

Additional comments

No Comments

---

## Round 0.2 · accepted · Accept

Dear DR. Uversky,
Thank you for your submission to PeerJ. I am writing to inform you that your manuscript, "Looking at microbial metabolism by high-resolution 2 H-NMR spectroscopy" (#2013:05:516:1:0:REVIEW), has been accepted for publication.